# Stem Cell-Derived Islets for Type 2 Diabetes

**DOI:** 10.3390/ijms23095099

**Published:** 2022-05-04

**Authors:** Andrew Salib, Fritz Cayabyab, Eiji Yoshihara

**Affiliations:** 1Lundquist Institute for Biomedical Innovation, Harbor-UCLA Medical Center, Torrance, CA 90502, USA; andrewsalib1999@gmail.com (A.S.); fritz.cayabyab@lundquist.org (F.C.); 2David Geffen School of Medicine, University of California, Los Angeles, CA 90095, USA

**Keywords:** diabetes, stem cells, human islet-like organoids, IAPP, glucolipotoxicity, disease modeling

## Abstract

Since the discovery of insulin a century ago, insulin injection has been a primary treatment for both type 1 (T1D) and type 2 diabetes (T2D). T2D is a complicated disea se that is triggered by the dysfunction of insulin-producing β cells and insulin resistance in peripheral tissues. Insulin injection partially compensates for the role of endogenous insulin which promotes glucose uptake, lipid synthesis and organ growth. However, lacking the continuous, rapid, and accurate glucose regulation by endogenous functional β cells, the current insulin injection therapy is unable to treat the root causes of the disease. Thus, new technologies such as human pluripotent stem cell (hPSC)-derived islets are needed for both identifying the key molecular and genetic causes of T2D and for achieving a long-term treatment. This perspective review will provide insight into the efficacy of hPSC-derived human islets for treating and understanding T2D. We discuss the evidence that β cells should be the primary target for T2D treatment, the use of stem cells for the modeling of T2D and the potential use of hPSC-derived islet transplantation for treating T2D.

## 1. Introduction

Diabetes encompasses metabolic disorders that lead to hyperglycemia and is caused by various factors such as infection, autoimmunity, obesity, and stresses [1,2,3]. Diabetes-induced hyperglycemia is linked with long-lasting co-morbidities such as hypertension, hyperlipidemia, chronic kidney disease, cardiovascular disease, retinopathy, and neuropathy [4,5]. As such, diabetes poses a significant global health risk with over 537 million individuals currently suffering from the disease and this led to 6.7 million deaths in 2021. According to the International Federation of Diabetes Atlas, the incidence of diabetes is projected to increase to over 783 million diabetic individuals by 2045 [3]. Diabetes is generally classified as either type 1 diabetes (T1D) or type 2 diabetes (T2D). Although both conditions result in a hyperglycemic condition, their etiology is markedly different.

T1D is characterized by an immune assault on the islet β cells [6]. The mechanism for T1D pathogenesis is highly heterogeneous but not fully understood yet. Multiple genetic and environmental factors can act as potential causes for immune autoreactivity against islet β cells. For instance, HLA-haplotype polymorphisms are involved in T1D susceptibility by influencing β cell interaction with immune cell populations. Variation in the INS gene can result in reduced insulin production in the thymus, thus preventing the detection and elimination of auto-reactive T-cells against β cells [7]. CFTR variants that hamper pancreatic exocrine function can result in increased inflammation, and thus greater T1D risk [8]. Additionally, changes in the microbiota, environmental conditions, or exposure to viruses or germs that exhibit molecular mimicry with endogenous factors of the body can trigger the production of autoantibodies against insulin and β cells [7].

Pancreatic islets isolated from T1D patients exhibit increased markers of ER stress as well as senescence and inflammatory signals, which lead to reduced expression of β cell identity markers such as insulin. The ER stress also disrupts the insulin:proinsulin ratio, which decreases the efficacy of glycemic control by insulin. This dysfunction is linked with the reduced β cell population and increased α cell population in T1D islets, leading to the dysregulation of glucagon secretion and accelerated blood sugar elevation [9,10].

T2D is by far the most common form of diabetes [11]. Modern diet and lifestyle habits have played an important role in exacerbating the prevalence of T2D [12,13]. T2D is becoming exceedingly burdening on the global healthcare systems with over 966 billion dollars spent [3]. The mechanism of T2D is not primarily involved with immune autoreactivity, but rather, β cell dysfunction and oxidative stress-induced inflammatory death in response to increased insulin resistance.

Diabetes treatment typically relies on insulin or insulin analog injections for maintaining normoglycemia. Although islet β cell dysfunction is considered the primary cause of hyperglycemia in both T1D and T2D, the recent advances in hPSC-derived islet systems, are currently being investigated mainly for T1D. hPSCs such as embryonic pluripotent stem cells (hESCs) [14] or human induced pluripotent stem cells (hiPSCs) [15] provide the opportunity to recapture human islet organogenesis and neogenesis in a scalable manner. In this review, we provide a perspective regarding the potential utilization of hPSC-derived islets in future studies on T2D.

## 2. T2D Pathogenesis: From Insulin Resistance to β Cell Death

T2D is a progressive disease that can be broken down into four stages: (1) defect in β cell glucose-stimulated insulin secretion (GSIS); (2) peripheral insulin resistance; (3) β cell compensation; (4) β cell loss. In this section, we discuss the progression of T2D and the potential advances in hPSC-derived islets (Figure 1).

### 2.1. Defect of GSIS

In healthy individuals, insulin secretion from functional β cells is an acute phase response. In the first phase, functional β cells secrete insulin within a few minutes of glucose stimulation followed by the second phase insulin secretion, in which a slower response is induced by continuous glucose stimulation. It was found that acute insulin secretory responses (AIR) to intravenous glucose are lower in individuals with impaired glucose tolerance and those at high risk of developing T2D [16]. Defects in the first phase of GSIS lead to inefficient glucose uptake in peripheral tissues and suppression of gluconeogenesis in the liver. Currently, glucagon-like peptide-1 (GLP-1) is one of the most popular targets for T2D treatment, as it is considered to enhance GSIS function in patients with glucose intolerance or T2D [17].

### 2.2. Insulin Resistance

T2D onset and progression is promoted by insulin resistance in peripheral tissues, particularly in the adipose, skeletal muscle, and hepatocyte tissues, and is often accompanied by obesity and related lipid dysregulation. Obesity can trigger chronic inflammation by recruiting macrophages in adipose tissues. Inflammation caused by macrophages, characterized by enhanced expression of cytokine markers such as nuclear factor kappa B (NF-κB) and tumor necrosis factor-alpha (TNF-α), induces insulin receptor resistance in adipocytes. Insulin is a strong suppressor of adipose lipolysis, therefore, defects of the insulin signaling in insulin resistance enhance lipolysis [18]. Partial inhibition of lipolysis through the inhibition of hormone-sensitive lipase has been found to result in a decrease in fatty acid influx into white adipose tissues, and an increase in glucose metabolism, lipogenesis, and insulin-mediated glucose uptake [19]. Insulin and fibroblast growth factor-1 (FGF-1) suppress lipolysis through phosphodiesterase 3B (PDE3B) and PDE4D, respectively [20]. Through inter-organ cross-talk, increased lipolysis at adipose tissues plays an important role in not only adipose insulin resistance but also the liver and skeletal muscle. Adipose lipolysis generates excess free fatty acids (FFAs). In 1985, Marchand-Brustel et al. showed that phosphorylation activity of the insulin receptor substrate 1 (IRS-1) in skeletal muscles was markedly reduced in obesity-induced diabetic mice [21]. Later, Griffin et al. showed that an increase in plasma-free fatty acids as a result of the infusion of lipids to mice undergoing a hyperinsulinemic-euglycemic clamp resulted in a significantly decreased glucose infusion rate [22]. The data also showed a significant decrease in skeletal insulin receptor signaling through attenuated tyrosine phosphorylation at insulin receptor substrate-1 (IRS-1) and AKT, which regulates Glut4 cycling and glucose uptake; further, a four-fold increase in protein kinase theta (PKC-θ) activity was observed. A follow-up study demonstrated that PKC-θ knockout mice did not exhibit such a decrease in insulin receptor activity and glucose uptake in response to fat accumulation within skeletal muscles [23]. Increased cytosolic diacylglycerols were observed upon post-lipid infusion induced PKC-θ mediated phosphorylation of serine residue 1101 on IRS-1, which inhibits the receptors’ tyrosine kinases [24]. Further studies have shown that lipids can also alter mitochondrial dynamics resulting in insulin insensitivity caused by lipid-induced mitochondrial fission [25]. A similar mechanism of insulin resistance has also been demonstrated in the liver, wherein increased diglycerides (DAGs) in response to lipid accumulation activate a different isoform of protein kinase C epsilon (PKCε), which in turn phosphorylates serine residues on IRS-2 and reduces glucose production and release in response to insulin signaling [26]. Thus, multi-organ insulin resistance increases the demand for insulin from β cells.

### 2.3. β Cell Compensation

Although defective GSIS and insulin insensitivity mark the initial stages of progression into T2D, patients can still maintain normoglycemia as long as the β cells exhibit adaptive functionality [27]. In response to insulin demand, β cells undergo proliferation as a compensatory defense mechanism against insulin resistance [28,29]. β cell function is known to exhibit a hyperbolic compensatory relationship with respect to insulin sensitivity, meaning that a decrease in insulin responsiveness within peripheral tissues is accompanied by a greater β cell insulin response. Patients that maintain this hyperbolic relationship can remain within normal glycemic levels. Patients whose β cells fail to maintain this compensatory relationship have been shown to develop glucose intolerance and ultimately T2D hyperglycemia [30]. Several studies have shown that a high-fat diet (HFD) induces glucose intolerance accompanied by insulin resistance, while still maintaining normoglycemia with an increased β cell mass and total insulin levels [31,32]. This β cell compensatory machinery in response to insulin resistance has also been observed in many prediabetic animal models [33,34]. However, the chronic hyperinsulinemia that results from the β cell compensation can, result in accelerated insulin resistance in peripheral tissues and eventually lead to β cell exhaustion and apoptosis.

### 2.4. β Cell Death

If the β cell compensatory mechanism fails, patients will no longer be able to maintain homeostatic glucose levels. Elevated postprandial glucose levels and glucose intolerance will eventually progress into full-blown diabetic hyperglycemia accompanied by β cell loss. Glucolipotoxicity, resulting from increased levels of plasma lipids and glucose has many other negative effects on β cells. We have previously identified that thioredoxin interacting protein (TXNIP), originally called Vitamin D3 upregulated protein (VDUP1) or thioredoxin binding protein-2 (TBP-2) plays a central role in β cell dysfunction and death [35,36,37,38,39]. TXNIP has a reciprocal function with the antioxidant protein Thioredoxin (TRX); thus, increased TXNIP expression is linked with enhanced oxidative stress [37,38,39]. TXNIP interacts with NLRP3, a component of the inflammasome to activate IL1β expression in islets, thus suggesting the mechanism of cytokine-induced β cell dysfunction in T2D [40]. Another leading hypothesis for β cell dysfunction is that the increased need for insulin production to counteract peripheral insulin resistances places exorbitant endoplasmic reticulum (ER) stress on β cells. Increased ER stress then activates the unfolded protein response (UPR) pathway, which in turn, can activate NF-κB to trigger β cell apoptosis [9,41,42]. ER stress was found to induce TXNIP expression; thus, TXNIP also plays a central role in ER stress-mediated inflammasome activation and β cell apoptosis in T2D [43]. ER stress also contributes to the initiation of islet amyloid polypeptide (IAPP) accumulation by stimulating protein misfolding in β cells, which further induces Chop-mediated cell death [44]. Masters et al. found that increased IAPP production in β cells activates the NLRP3 inflammasome, leading to inflammation and macrophage recruitment. Human IAPP (hIAPP) but not rat or mouse IAPP shows toxicity in β cells by an accumulation of hIAPP dimers and more complex amyloid oligomer formation, triggering an autophagic response to restrict β cell mass [45,46]. T2D is often accompanied by aging, with an enhanced glycolytic and reduced oxidative metabolic signature, which contributes to reducing glucose sensitivity in aged diabetic β cells, as insulin secretion is tightly regulated by metabolic function [47,48]. Aguayo-Mazzucato et al. further found that aging in conjunction with insulin resistance both contribute to β cells that exhibit increased markers of senescence. These senescent β cells are resistant to apoptosis and secrete senescence-associated secretory phenotype (SASP) factors such as inflammatory cytokines, proteases, and DNA fragments, which further induce dysfunction in the surrounding β cells. Increased senolysis through oral ABT263 has been found to improve β cell metabolic function and reduce T2D symptoms. Other metabolic inquiries such as those by Lupse et al. have found that PHLP phosphatase inhibition can protect β cells from diabetic death [49]. Hyperglycemic stress chronically activates the mTORC1 pathway, which in turn upregulates protein phosphatase PHLPP-like protein (PHLP) pathways. PHLP phosphatases will deactivate the cellular signals for β cell survival such as AKT which in turn will trigger β cell apoptosis and contribute to T2D development.

In addition to changes in metabolic function, apoptotic rate, insulin production and processing and inflammation, β cells have been shown to undergo dedifferentiation in response to hyperglycemic stress. Lineage tracing experiments in mice have revealed that β cells dedifferentiate into a progenitor-like state as exemplified by the upregulation of pluripotency markers, such as NANOG and OCT4 [50]. Furthermore, dedifferentiation often leads to the neogenesis of other hormonal cells including glucagon-producing α cells in T2D patients [51]. Sox5 knockdown significantly reduces GSIS and calcium-induced exocytotic depolarization, whereas Sox5 overexpression enhances the gene expression necessary for β cell identity, thus improving insulin secretion by β cells in mouse T2D models. Oxidative stress has been suggested as a leading factor for ectopic β cell fates, with increased oxidative mitochondrial stress being associated with upregulated JNK pathways and downregulated β cell identity genes such as FOXO1, MAFA, and PDX1 [52,53,54,55,56]. Oxidative stress in response to glucolipotoxicity has also been shown to result in increased ROS production through NADPH oxidase (NOX), which in turn impacts β cell failure through aberrant changes in the mTOR pathways that affect β cell insulin production, β cell mass, proliferation, dedifferentiation and mitochondria-mediated apoptosis [57,58].

T2D progression is accelerated by insulin resistance but is critically dependent on β cell failure. Therefore, it is important to understand the stage-specific event of β cells during T2D pathogenesis. Prior to β cell death, diet as well as exercise, supplemented with insulin sensitizers may reverse a prediabetic state with β cell restoration. However, once full T2D has manifested through β cell death, the most important long-term solution is β cell regeneration or replacement. This is supported by recent studies, which demonstrated that a decrease in visceral fat is only able to induce diabetic remission if β cells maintain a capacity for functional recovery [59]. Following these aspects, we suggest the hPSC-derived islets can be used for screening the drugs effectively restores β cells in the early stage of T2D, whereas transplantation of hPSC-derived islets is a new therapeutic for late-stage of T2D.

## 3. Disease Modeling Using hPSC-Derived Islets

hPSC-derived islets have opened a new avenue to study T2D pathogenesis. Difficulties in the acquisition, culture and long-term maintenance of primary human islets present a challenge in studying the genetic, as well as developmental factors contributing to T2D pathogenesis. Commonly used cell culture lines such as the human β cell line, EndoCBH1 cells [60], mouse β cell line, MIN-6 [61] or rat β cell line INS-1 cells [62] do not recapture the proper human β cell transcriptome and physiological properties because of their proliferative nature as cancer cell lines. To overcome these different physiological and genetic backgrounds compared to the non-cancerous counterparts, as well as the multi-cellular nature of endocrine human islets cells, we recently developed three dimensional (3D) structured functional human islet-like organoids (HILOs) from hPSCs, which are capable of secreting insulin in response to glucose [63]. These hPSC-derived islets present a unique tool to study the genetic basis of T2D and also to develop novel therapeutics through genetic engineering and high throughput drug screening that are simply not possible with direct experimentation on cell lines or primary islets. In this section, we will illustrate how recent developments in stem cell diabetes modeling reflect the potential advantages of the field.

Modern advances in genome-wide association studies (GWAS) have enabled researchers to uncover how loci variants can contribute to fat storage, lipid metabolism, and most importantly, genetic susceptibility to insulin resistance and β cell dysregulation. However, establishing experimentation to establish causal relationships between genetic variations and diabetic phenotypes remains challenging [64]. Combining genetic engineering techniques such as CRISPR-Cas9 with physiologically accurate hPSC-derived islets can enable elucidation of mechanisms underlying the monogenic and, in some cases, the polygenic causes of T2D [65,66]. There are two diametric ways of using hPSCs to study how singular genes can contribute to diabetes. The known genes responsible for contributing to T2D development can be mutated or knocked out by gene editing in hPSC-derived islets in order to study the time-temporal pathogenesis of diabetes. Similarly, using hiPSC technology, hiPSC-derived islets can be derived from differentiated cells of patients, with known mutant genes in order to create “personalized diabetes in a dish” models. Zeng et al. utilized CRISPR-Cas9 to create hESC lines with biallelic knockout (KO) mutations in genes such as CDKAL1, KCNQ1, and KCNJ11, which have been identified as significantly related to T2D through GWAS analysis [67]. CDKAL1KO hESC-derived β-like cells were more susceptible to high-glucose or palmitate-induced ER stress and apoptosis. Co-culture with high concentrations of glucose or palmitate exhibited increased apoptotic and ER stress markers, highlighting how this gene influences glucolipotoxicity responses. CDKAL1KO, KCNQ1KO, and KCNJ11KO hESC-derived β like cells show defective GSIS and impaired ability to maintain glucose homeostasis in STZ-induced diabetic NOD-SCID mice. These reverse genetics approaches in human disease models successfully identified novel mechanistic insights regarding the involvement of the metallothionein (MT) pathway as responsible for the defective function of CDKAL1KO hESC-derived β-like cells [68]. INS gene mutation in hiPSCs that were successfully differentiated into β-like cells revealed that INS mutation enhances ER stress signals accompanied by reduced Insulin-mTORC1 signaling [69]. Thus, giving us insights into how INS deficits can contribute to neonatal diabetes. hPSC-derived β-like cells can also be utilized to study the embryogenesis and development of pancreatic islets in vitro, allowing us to study how key developmental genes can impact islet generation and in turn cause congenital diabetes. The CRISPR/TALEN KO system has been utilized to study how pancreatic islet organogenesis is impacted by KOs in PDX1, RFX6, NGN3, HES1, ARX, MNX1, PTF1A and GLIS3 [70]. Many more studies have identified β-cell deficiencies caused by key developmental genes such as WFS1, HNF1A, GATA6, HNF4A, HNF1B, and STAT3 in the human context by using hPSCs technologies, and have been reviewed in detail elsewhere [66,71,72]. Human-patient-derived hiPSCs have become powerful tools to study the pathogenesis of diabetes and the development of possible gene correction therapy. Lithovius et al. found that iPSCs-derived from patients with a homozygous mutation in the sulfonylurea 1 receptor subunit of the K_ATP_ channel can be differentiated into β-like cells with aberrant insulin secretion at low glucose concentrations, accompanied by proliferation as compared to CRISPR corrected patient hiPSC-derived β-like cells [73].

While hPSC-derived β-like cells or islets are extremely versatile tools for studying β cell dysfunction, they can also provide us with the ability to study the genetic roots underlying the more complicated environmental stresses-induced T2D. Kim et al. showed that the autophagy enhancer (MSL-7) reduces hIAPP oligomer accumulation in hiPSC-derived β-like cells and diminishes oligomer-mediated apoptosis of β cells, which is mediated by transcription factor EB (TFEB) [45]. This in vitro humanized model recapitulated some part of hIAPP-induced ER stress and β-cell death in mice fed a high-fat diet, a model of T2D onset. Modeling human T2D, with environmental factors along with genetic mutations can be a great tool to study previously unknown mechanisms of disease progression. Together with gene-editing technology [74], targeting therapeutics to rescue the phenotype mimicked in vitro may hold the promise for treating T2D in the future.

## 4. Screening Assays for New Therapeutics

Human β cell differentiation is a complex and multistep process that requires a step-by-step differentiation of hPSCs directed into definitive endoderm (DE), then foregut (FG), pancreatic progenitor (PP), endocrine progenitor (EP), and insulin-producing β-like cells [75]. hPSC-derived β-like cells or islets can provide an efficient platform to assess the effects of novel drugs on β cell development, survival, and function. High-throughput chemical library screening of hiPSCs expressing a human insulin promoter-driven GFP and human-NGN3 promoter-driven mCherry (hINS-GFP/hNGN3-mCherry) dual reporter system found that a fibroblast growth factor receptor 1 inhibitor that promoted this terminal differentiation stage and enhanced GSIS [76]. Similarly, a large-scale chemical library (~23,000) was used to identify molecules that can promote the development of the definitive endoderm (DE) and pancreatic progenitor (PP) stage [5,77]. ROCK inhibitor was identified as an important factor for directing hPSC differentiation to PP as demonstrated by upregulated FOXA2 and PDX1+ gene expressions [78]. Potent ATP-competitive inhibitors of Akt1/2/3 and p70S6K/PKA were identified to enhance PP proliferation and differentiation [5]. Using high-throughput screening to identify signaling molecules that can maintain the proliferative capacity of intermediate progenitor stages can significantly reduce the time, effort, and cost associated with human β cell production Further, it may help minimize the genetic heterogeneity due to different stem cell origins of differentiated β cells, thus minimizing the variability in β cell differentiation and functional maturation. PP cultured on 3T3-J2 feeder cells using a combination of EGF, FGF10, and retinoic acid activation and Notch and TGF-β inhibition enabled >25 passages of human PPs [79]. Identifying novel signals of pancreatic lineage specification can also contribute to developing new ways to expand PPs for further scaling human β-like cell generation.

CDKAL1KO hESC-derived β-like cells were used to identify the small molecules that could rescue the lipotoxicity-induced β-like cell dysfunction [67]. Over 2000 small molecules were tested, and FOS/JUN pathway inhibitors were found to improve insulin secretion and β-like cells survival. We and colleagues have utilized hiPSC-derived β-like cells labeled with human insulin promoter-driven GFP for screening the essential genes for β cell identity. Doxycycline (dox)-inducible Cas9 induction together with single guide (sgRNA) libraries have enabled us to identify vitamin D receptor and bromodomain-containing proteins (BRDs) as novel genes that contribute to maintaining the identity of human β-like cells. [80]. Further investigation revealed the physiological interaction between VDR and BRD7 or BRD9, and the novel mechanism of PBAF/BAF mediated anti-inflammatory responses. Interleukin-1β (IL-1β)-mediated cytokine stress-induced β cell dedifferentiation and cell death are synergistically protected by VDR agonists and BRD9 inhibitors. This study provided a proof-of-concept that CRISPR screening in hiPSC-derived β-like cells can reveal a new pathway to treat T1D and T2D.

Further expansion of these screening approaches will uncover new drugs that can be more effectively utilized for treating heterogenic T2D and to improve our capability to generate unlimited functional human β cells (Table 1).

## 5. Modeling Organ–Organ Interactions Using hPSC-Derived Tissues

While iPSC- and hESC-derived β-like cells alone can provide us with an important model to interrogate the basis of T2D, it is important to acknowledge the fact that many complicated organ failures involve β cell dysfunction and T2D progression. β cell functionality is influenced by many complex organ-organ hormonal communications such as hepatokines from the liver, incretins production from the intestines, osteocalcins from bones, leptin from adipose tissues, myokines from skeletal muscles, and cholinergic parasympathetic stimulation from the autonomic nervous system [99]. Combining hPSC-derived peripheral tissues and hiPSC-derived β-like cells can provide better insights into the interplay between inter-organ relationships, insulin resistance, and β cell dysregulation. Furthermore, modern advances in microfluidic devices and organ-on-a-chip technology can allow us to study these interactions on an efficiently fine-tuned scale [100]. For this purpose, disease modeled hPSC-derived adipose, liver, peripheral nerves, and muscle cells combined with hiPSC-derived β-like cells will be useful to grasp better insights into the development of T2D accompanied by insulin resistance caused by cellular or tissue autonomous genetic defects, systemic factors such as hormonal changes, hyperinsulinemia, and chronically elevated lipid accumulation in visceral, skeletal and adipose tissues.

hPSC-derived peripheral tissues can also be utilized to study the comorbidities of diabetes. Several groups have obtained skeletal muscle biopsies and cultured primary myoblast cultures from patients with T2D and controlled diabetes, and from healthy control patients [101,102]. These primary myoblast cultures were reprogrammed into hiPSCs for creating hiPSC-derived myoblast cultures (iMyos) from control and T2D patients. hiPSC-derived myoblasts rather than primary myoblasts derived from healthy control and T2D patients were used as hiPSC reprogramming minimizes epigenetic differences between the studied cultures and normalizes extracellular environmental differences. hiPSCs have a potentially unlimited capacity for in vitro culture. This prevents the need for recruiting new patients and the need for increased muscle biopsies. The T2D iMyos revealed that insulin resistance can be recaptured using this in vitro muscle T2D model. Defective insulin signaling for glucose uptake is affected by phosphorylation response in PI3K/AKT, GSK-3αS21, GSK-3βS9, FOXO1T24, and FOXO3aT32, as well as increased basal phosphorylation of IRS-1 (at S1101), IRS-1 (at S1078), IRS-2 (at S770), and IRS-2 (at S779) in the iMyos. In addition, to insulin regulatory and secretory networks, other basic cellular networks, such as mRNA metabolism and splicing, vesicle-mediated transport, apoptotic cleavage of proteins, and cell cycle controlling signals were also disrupted in T2D iMyos. Human adipocyte-derived stem cells (hADSCs) also provide a powerful tool through which to study how insulin resistance manifests in human adipose tissues. hADSCs were utilized to create 3D fat tissues in physiological-like systems that exhibit hormonally sensitive functions [103]. Such 3D white adipose tissue (WAT) models can allow us to genetically and chemical assay for insulin sensitizers and desensitizers [104,105]. These WAT models, as well as the recently developed hPSC-derived brown fat (BAT) models, provide better tools to understand human fat development and physiology [106]. In addition to adipose-derived tissues and muscle-derived tissues, recent advances have allowed the generation of functional vascularized liver buds or hepatocyte organoids that can be used to model hepatic steatosis as well as for disease modeling, detoxification of drugs, regeneration, and production of serum albumin [107,108,109,110,111,112,113]. iPSC lines have been generated from urine epithelial cells of T2D patients [114]. These T2D hiPSC cell lines were then differentiated into cardiomyocytes to investigate the cellular phenotypes in the T2D background. As expected, the cellular phenotypes of T2D-induced cardiomyopathy such as cardiac hypertrophy, intracellular lipid accumulation, apoptosis in response to high glucose and fat conditions, as well as abnormal electrical signaling and arrhythmias were observed. These hiPSC-derived cardiomyocytes underwent transcriptomic analysis and were found to have upregulated TGF-β signaling. The inhibition of TGF-β signaling helped prevent hypertrophic cell growth. Although, hiPSC-derived cardiomyocytes enable researchers to gain a better molecular understanding of diabetes-induced physiological, cellular, and signaling changes in heart failure, the complication of heart failure is not solely contributed to by the cardiomyocytes. Therefore, to model more complex disease progression and molecular mechanisms, new systems are needed to study simultaneous multi-organ functions.

The field is moving forward to study cell–cell or organ–organ interaction by connecting individual differentiated cells. This idea has been mainly tested in the organ-on-a-chips system [100] or more recently by creating intermediate organs through the fusion of 2 or more differential cell types [108,115]. Microfluidics technology has been used to enhance human islet survival and functionality [116]. Besides these advances, a hepatocyte-pancreatic islet co-culture model on a microfluidic biochip was found to recapture physiological islet–liver interaction as a part of T2D progression [117]. Co-culturing of primary human islets and primary hepatocyte spheroids in a microfluidics biochip showed that functional coupling between islets and the liver synergistically regulates insulin secretion and glucose uptake. Such technologies can be co-opted with hPSC-derived islets to examine even more biologically applicable and relevant organ–organ interactions. For instance, Gesmundo et al. demonstrated that exosomes derived from adipose tissues play an important role in the transcriptional and metabolic profile of β cells. The exosomes from obese-derived adipocytes cause greater cytokine inflammation and β cell deficiencies. [118]. Multi-organ chips can be developed using co-culture hPSC-derived islets with inflamed adipocytes, insulin-resistant skeletal, and hepatocyte tissues to gain a better understanding of how hormonal and molecular communication between the two classes of tissues influences β cell dynamics. The addition of glucose and fatty acids in the model can allow even more complex and interactive models that can reveal how peripheral tissues respond to insulin secretion by hPSC-derived islets and whether increased glucose and lipid levels as a result of insulin resistance can mimic in vivo glucolipotoxicity on microfluidic multi-organ chip devices.

## 6. Islet Transplantation in T2D

One of the most important yet challenging use cases of hPSC-derived islets is their capacity to be used as a treatment and long-term solution for insulin-dependent diabetes. Recently, advances in β cell differentiation from iPSCs and hESCs have allowed the development of glucose- and hormone-sensitive human islets in vitro [63,119,120,121,122]. This idea has been explored in both pre-clinical models [63,119,120,121,122] and clinical trials [123]. Ramzy et al. and Shapiro et al. analyzed the results of a phase 1/2 clinical trial, respectively (NCT03163511) [123,124]. In this clinical trial, T1D patients were implanted subcutaneously with a macro encapsulation device containing hESC-derived PPs. After one year of implantation, along with immunosuppressive treatment, the grafts were removed and were found to have increased vascularization. Subsequent immunohistochemical analyses validated the markers for INSULIN and MAFA, denoting successful in vivo differentiation into endocrine β cells. The patients also exhibited elevated levels of human c-peptide following meals and a reduced need for exogenous insulin injections. Several issues were identified, such as adverse effects in the patients as a result of the immunosuppressant treatment as well as poorer glycemic control as a result of the asymmetric differentiation of hESC-derived PPs into glucagon-producing α-like cells rather than β-like cells. However, these studies have encouraged the safe use of hPSC-derived islet cell therapy in T1D patients.

We and others have recently shown the possibility to engineer more immune-tolerant hPSC-derived islets for cell therapy in T1D models, thus requiring less immunosuppressants [63,125,126]. We have discussed further advances in immune evasive and immune tolerance therapies as well as the ideal transplantation sites of transplantation elsewhere [127,128]. Islet transplantation treatment research is primarily focused on treating T1D, while it may also be effective for late-stage T2D with loss of β cell function and mass. Although caused by different initiating factors (autoimmunity vs glucolipotoxicity as a result of insulin resistance), T2D is also critically dependent on β cell dysfunction, and apoptosis [129,130,131,132,133,134,135,136,137]. Thus, it may be possible to expand the current efforts to use hPSC-derived islets for treating T2D as well as T1D (Table 2).

Earlier studies have tested this idea by transplanting isogenic mouse iPSCs-derived insulin-producing cells through the intraportal vein in T2D models db/db mice [138]. The results showed long-term glucose normalization and increased levels of in vivo insulin were observed, suggesting that as a proof-of-concept, hPSCs-derived islet therapy is effective in treating T2D conditions. Further analysis revealed that a few mice failed to maintain normoglycemia and that some of the mice became less responsive to exogenous insulin injections indicating the re-establishment of insulin resistance. Aside from universal concerns regarding immuno-reactivity for any type of transplanted non-autologous material, the eventual dysfunction of transplanted hPSC-derived β-like cells because of prolonged glucolipotoxicity, as well as IAPP deposit toxicity from the continued insulin resistance in peripheral tissues, represents the main challenge for β cell transplants in T2D. Since insulin resistance plays an important role in modulating β cell functionality and survival, combination therapies that improve insulin sensitivity, glucose exclusion, or reduce gluconeogenesis together with islet transplantation may be required for ensuring that transplanted β cell populations remain healthy and functional. Combination therapies may include sodium/glucose cotransporter 2 inhibition (SGLT-2i) therapy [143], GLP-1 agonism/DPP-4 inhibition [144], metformin [145] and PPAR agonism/Thiazolidinediones [146]. Using another cell product such as human adipose-derived stem cells (hADSCs) or hPSC-derived BATs together with hPSC-derived islets to enhance insulin sensitivity or metabolic activity may also be worthwhile. Autologous injections of mADSCs into T2D model mice have been found to result in increased insulin sensitivity as marked by a decrease in glucose plasma levels, a decrease in adipocyte size, a restoration of β cell mass, and a reduction in pro-inflammatory insulin desensitization markers, such as IL-6, TNF-α, and F4/80 in the liver. Mice transplanted with BATs by PRDM16-mediated direct conversion from hiPSCs maintained lower weight gain in response to a high-fat diet, reduced basal glucose levels, and reduced plasma levels of LDLs, and free fatty acids as compared with those in control mice. These approaches to enhancing insulin sensitivity using cell therapy have the potential for treating T2D [147,148,149,150].

More studies are needed to test the idea that hPSC-derived islet cell therapy in T2D may eventually provide a functional cure for T2D.

## 7. Conclusions

Currently, there is no functional cure therapy available for human T2D. Owing to the complexity of human T2D, spatiotemporal human cellular models that integrate progressive multi-organ failure are necessary to recapture the fundamental mechanisms and therapeutic targets for T2D. Recent advances in hPSC-derived β-like cells or islets sophisticatedly designed to be functional and more physiologically relevant have facilitated studies on the molecular basis of β cell failure and possible new therapeutics to cure human T2D. These approaches include drug screening, gene screening, and islet transplantation, and can be expanded with new findings. More studies are needed to examine the utility of hPSC-derived β-like cells or islets in a pre-clinical setting with regard to the human relevance of this novel targeted therapeutics for T2D.

## Figures and Tables

**Figure 1 ijms-23-05099-f001:**
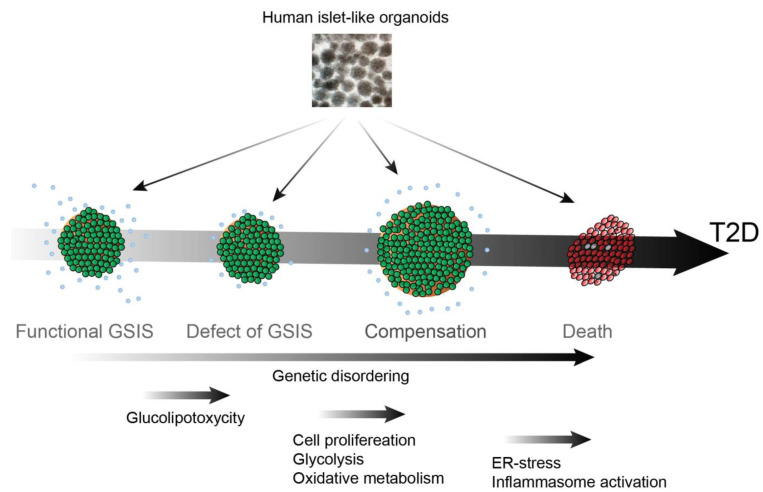
hPSC-derived islet for studying the pathogenesis of T2D.

**Table 1 ijms-23-05099-t001:** Screening approach by using hPSC-derived β-like cells or islets β-like cells identified target.

Description	Cell Resource	Screening	Screened Phenotype	Reference
VDR, BRD7 as epigenetic modifiers of β-cell anti-inflammation	hiPSC-derived β-like cells	CRISPR-Cas9:Gecko library	Reduced Insulin promoter-driven GFP expression.	[80]
Fibroblast growth factor receptor 1 inhibitor (PD166866) for pancreatic progenitor differentiation into β-like cells	hiPSC derived pancreatic progenitors to terminally differentiated islets	Chemical library (Lopac-pfizer)	Dual Insulin promoter-driven GFP expression and hNGN3 promoter-driven mcherry expression	[76]
Rock inhibition (Rocki) (Fasudil and RKI-1447) for hPSC differentiation to pancreatic progenitor cells	mESCs and hESCs (H9ES) to PDX1+ pancreatic progenitors	Chemical library (23,406 small molecules)	FoxA2 driven-Venus reporter followed by qPCR and FACS analysis of PDX1	[77]
ATP-competitive inhibitor of Akt1/2/3 and p70S6K/PKA (AT7867) for pancreatic progenitor in vitro proliferation	hiPSC-derived PDX1+ pancreatic progenitors	Chemical library (Custom small molecule library comprising kinase inhibitors)	Cell number readout of immunohistochemically stained nuclei followed by immunostaining of Ki67 and PDX1	[5]
FOS/JUN inhibition (T5224) for β-cell protection against glucolipotoxicity in T2D	CDKAL1KO hESC (HUES3)-derived β-like cells	Chemical library (2000 different drugs based on FDA-approved drugs and clinical trial candidate drugs)	Ratio of propidium iodide-stained dead cells and insulin promoter-driven GFP expressing cells.	[67]
miRNA-690 as RNA regulator of stem cell differentiation into β-like cell	miPSCs-derived β-like cells	miRNA microarray assay	RT-qPCR analysis of miRNA samples to identify differentially expressed miRNAs in differentiated β-like cells	[81]
CD26- andCD49A+ for capturing β-cell enriched hESC-derived islets	hESC (HADC-100)-derived islets	Functional Cell-Capture Screening using 235 antibodies that bind to cell surface proteins	Amount of Insulin expressing cells as measured using anti-Insulin antibodies (IHC)	[82]
Platelet-derived growth factor receptor (PDGFR) and kinase inhibitor (Tyrphostin9) for improved β-like differentiation from hiPSCs	hiPSC derived β-like cells	Chemical library using 80 kinase inhibitors and 43 WNT signaling modulators	PDX1 promoter-driven mcherry expression	[83]
Novel synthesized antibody clones 4-2B2, 4-5C8, and 4-5G9 for capturing stem cell-derived islets enriched in mature β-like insulin-producing cells	hESC (MEL-1)-derived β-like cells	1248 antibody hybridoma clones	FACS sorting of hybridoma clones capturing high number of insulin promoter-driven GFP fluorescent cells	[84]
Proof of Concept (NGN3, GATA4, GATA6, TET1, TET2, TET3, PDX1, RFX1, PTF1A, GLIS3, MNX1, HES1, ARX) for lineage determinants of pancreatic progenitor development and β- cell differentiation	hESCs (HUES8, HUES9, MEL-1) hiPSC	iCRISPR (TALEN and Doxycycline-inducible CRISPR/Cas9 System)		[70,85,86]
Identification of intestinal organoids capable of converting intestinal crypt cells into endocrine cells	hESCs (H1ES, H9ES)-derived intestinal organoids	Transduction of Pdx1, Mafa and Ngn3 using a lentiviral system	Pdx1, MafA, Ngn3 expression with GFP reporter and RNA sequencing of INS1 and SUR1 transcriptional expression	[87]
ROCKII as regulator of β-cell maturation	hESC (HUES8) differentiated into pancreatic progenitor population containing more than 85% PDX1^+^ cells	LOPAC library and MicroSource Spectrum Libraries	INS+ cells via insulin antibody staining	[88]
(−) Indolactam V induces generation of pancreatic progenitors from definitive endoderm	hESC (HUES9)-derived endoderm cells	High-Content Chemical Screening (Sigma LOPAC libraries, MicroSource US-Drug collection and Prestwick Chemical library	Pdx1+ cells	[89]
Staupirimide inhibits nuclear localization of NME2 that leads to downregulation of c-Myc, a key regulator of pluripotent states, allowing for priming of hESC for efficient differentiation	hESC (H1ES) differentiated into definitive endoderm	Approximately 20,000 compounds corresponding to diverse chemical scaffold from a kinase-oriented library generated in house	Sox17^+^ cells versus total DAPI^+^ nuclei	[90]
TGFβ activators IDE1 and IDE2 induce differentiation of ESC towards endodermal lineage	hESCs (HUES4, HUES8 and HUES9)	MicroSource Library and HDAC-inhibition based on synthetic, bioactive and natural compounds (Stuart L. Schreiber Lab)	Sox17 promoter-driven dTomato reporter	[91]
RNLS for β cell protection from immune attack	NIT-1 β-cell line, confirmed by hESC (HUES8)-derived β-like cells	CRISPR Gecko library of 60,000 gRNAs comprising over 19,000 genes	Screening of β-cell survival after splenocyte induced killing of β-cells in transplanted NOD-scid mice	[92]
Galunisertib activates TGFβ signaling that rescues GLIS3^−/−^ associated diabetes	GLIS3^−/−^ hESC (HUES3)	In-house library of ~300 signaling modulators from an epigenetics library (Cayman Chemical), Prestwick library of approved drugs (FDA, EMA, and other agencies), LOPAC (Sigma Aldrich) and the MicroSource library totaling ~5000 chemicals	Staining with anti-insulin antibodies and anti-cleaved caspase 3 antibodies	[93]
Role of mTORC1 activity in functional shift from amino-acid responsive to glucose-responsive insulin secretion, demonstrating the role of mTORC1 in the initiation of functional maturation of pancreatic β-cells	hESC (HUES8)-derived β-like cells	Amino acid stimulation	Single-cell RNA seq of fetal human islets to identify signaling pathways correlated with β-cell differential responses to varying nutrients and FACS-based assay to quantify mTORC1 activation	[94]
Jun N-terminal kinases (JNK)-JUN family genes that co-occupy ESC enhancers with OCT4, NANOG, SMAD2, and SMAD3 which prevent exit from pluripotent state, exemplifying their barrier function for definitive endoderm differentiation	iCas9 hESC (HUES8)	Pooled lentiviral human Gecko v2 library consisting of 58,028 gRNAs targeting 19,009 genes (3 gRNAs per gene)	GFP reporter of Sox17	[95]
Propargite, a commonly used pesticide induces β-cell death	Direct differentiation of hESCs (H1ES, H3ES) into isogenic β-like cells	Phase I Toxicity forecaster (ToxCast) Library	Staining with anti-insulin antibody	[96]
Harmine and INDY function promote adult β cell proliferation, targeting dual-specificity tyrosine-regulated kinase-1a (DYRK1A) as a target. Inhibition of DYRK1A, SMAD and trithorax induce robust replication of hPSC-derived β-like cells and adult human pancreatic β-cells	INS1 and βTC3 cell lines, validated in Mel1 hESC (MEL-1)-derived β like-cells	2300 compounds from MicroSource Discovery System and 100,000 compounds from Chembridge	Luciferase-based high throughput screening	[97,98]

**Table 2 ijms-23-05099-t002:** Summary of β cell transplantation studies for T2D Highlights.

Description	Donor Islet (Source and Type)	Recipient Animal Model of Type 2 Diabetes	Transplantation Location	Reference
All mice displayed hyperglycemia. Post implantation all 30 fully engrafted mice displayed homeostatic normoglycemia. Three control mice were engrafted with non-insulin control cells and maintained hyperglycemia. After 12 weeks, of the 15 mice that were not sacrificed for histology and that survived surgical complications, two re-developed hyperglycemic insulin resistance and the remainder maintained proper glucose homeostasis	200,000 iPS derived insulin-secreting β-like cells that were enriched for insulin expression from initial pool of differentiated cells by FACS sorting	T2D mouse model (*Lepr^db^*, C57BLKS; *Dock7^m^*, DBA/J)	Intraportal vein injection	[138]
Serum insulin concentration was higher in the CD154 and tacrolimus co-administered group, compared to the db/db group after 3 days post-transplantation. The grafted islets were detected 14 days post-transplantation via immunohistochemistry	500 Islet equivalent from Sprague-Dawley male rats (age of 8 weeks), surface camouflaged with 6-arm-PEG-catechol	db/db C57BL/KsJ male diabetic mice, and db/db mice co-administered with anti-CD154 antibody and tacrolimus	Kidney capsule	[139]
Transplanted islets readily engrafted onto the iris and became vascularized. Progression of diabetes was reversed, with significant decrease in fasting glucose observed while graft was in place. Metabolic markers, hemoglobin A1c and fructosamine showed improvement after transplantation. No changes in intraocular pressure, cataract formation, ophthalmitis, or retinal vessel deformation were observed	1500 allogeneic donor islet equivalent/Kg	Nonhuman primate model of T2D (cynomolgus monkey with high-fat-diet-induced T2D)	Anterior chamber of one eye	[140]
Transplanted mice showed reduced serum glucose level to 200mg/dL at 6 weeks post-transplantation and improved reduction in glucose level during intravenous glucose tolerance test. Furthermore, transplanted mice have lower HOMA-IR and higher Matsuda Index	400 isogenic islets from eight-week old male C57BL/6N	Eight-week-old male C57BL/6N fed with high-fat diet for 4 weeks, before being intraperitoneally injected with low-dose streptozotocin twice within 24 h	Kidney capsule	[141]
Smad3KO islets transplantation produced lower serum glucose level and lower hemoglobin A1c compared to WT islet transplantation, in both T1D and T2D murine models. Furthermore, Smad3KO islet transplanted models have better kidney function compared to WT transplanted models	250 Islets from 4-12 week old C57BL/6 or Smad3KO	db/db male mice with in C57BLKSbackground	Kidney capsule	[142]

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
