# Peer review of "Stem Cell-Derived Islets for Type 2 Diabetes"

_ijms, 2022, doi:10.3390/ijms23095099_

Round 1
Reviewer 1 Report
Dear authors,
I congratulate you for the article that intends to review Stem cell-derived islets for Type 2 diabetes.
The topic of the article is very interesting for the specialist public in the field. However, when using a review it is very important that it follows the criteria established in the journal for this type of study.
Please adapt the article to the magazine's recommendations for revisions.
I await the modification of the article.
Author Response
Thank you for your instructive comments for our manuscript. We have corrected the structure of Figure. 1, Table.1 and Table.2, moved to the after conclusion section following IJMS template. In addition, we have discussed with Editor and we confirmed they will be in charge for the layout edition before production. Minor English collection will be also performed by Editor (Accroding Ms. Aisa Safaya).
All revised sentences and structures are highlighted by track records.
Reviewer 2 Report
The review of Salib et al. is dealing with the use of new cellular technologies, human pluripotent stem cell (hPSC)-derived human islets in the managment of diabetes. Although the topic is hot I have some concerns.
The paragragh dealing with T2D pathogenesis is too long for a paper which topic is mainly focused on stem-derived islet for the therapy and pathophysiology of diabetes. So it should be deleted or substantially shortened. Also lines 84-87 have a criptic meaning.
Tables (1 and 2) make the text difficult to read so they should be in supplementary or otherwise structured differently to improve reading.
Author Response
Thank you for your thoughtful reviews to our manuscript. We have corrected the structure of Figure. 1, Table.1 and Table.2, moved to the after conclusion section following IJMS template. We have re-written lines 84-87 “Obesity can trigger chronic inflammation by recruiting macrophages in adipose tissues. Inflammation caused by macrophage, characterized with enhanced expression of cytokine markers such as nuclear factor kappa B (NF-κB) and tumor necrosis factor-alpha (TNF-α) induce insulin receptor resistance in adipocytes. Insulin is strong suppressor of adipose lipolysis, therefore defect of the insulin signaling in insulin resistance enhances lipolysis [18].“
Our current manuscript is mainly focus on the utility of stem-derived islet in T2D study. Our previous study (Yoshihara et al, Nature, 2020) demonstrated that the efficacy of stem-derived islet in T1D and now the field is also interested in exploring the utility of stem-derived islet in T2D. However, less study has successfully integrated this idea due to the complexity of T2D pathogenesis. Our purpose of this study is the dissecting the event in islets during T2D pathogenesis for spatiotemporal T2D modeling of stem-derived islet and identifying therapeutic targets. Therefore, the detailed statement about T2D pathogenesis is necessary. To reduce the stress of reading about the paragragh dealing with T2D pathogenesis, we have added citation of Figure 1, which is the scheme of T2D pathogenesis and it’s utility of stem cell derived islets in the front summary section of the paragraph.
In addition, we have discussed with Editor and they will be in charge for the layout edition before production. Minor English collection will be also performed by Editor (According Ms.Aisa Safaya).
All revised sentences and structures are highlighted by track records.
Round 2
Reviewer 2 Report
The manuscript gained in readability.